# P-Cadherin Is Expressed by Epithelial Progenitor Cells and Melanocytes in the Human Corneal Limbus

**DOI:** 10.3390/cells11121975

**Published:** 2022-06-20

**Authors:** Naresh Polisetti, Lyne Sharaf, Gottfried Martin, Günther Schlunck, Thomas Reinhard

**Affiliations:** Eye Center, Medical Center, Faculty of Medicine, University of Freiburg, Killianstrasse 5, 79106 Freiburg, Germany; lyne.sharaf@uniklinik-freiburg.de (L.S.); gottfried.martin@uniklinik-freiburg.de (G.M.); guenther.schlunck@uniklinik-freiburg.de (G.S.); thomas.reinhard@uniklinik-freiburg.de (T.R.)

**Keywords:** limbal stem cells, limbal niche cells, mesenchymal stem cells, melanocytes, limbal epithelial progenitor cells, corneal tissue engineering, cadherins, P-cadherin, cell–cell interactions, limbal stem cell niche

## Abstract

Interactions between limbal epithelial progenitor cells (LEPC) and surrounding niche cells, which include limbal mesenchymal stromal cells (LMSC) and melanocytes (LM), are essential for the maintenance of the limbal stem cell niche required for a transparent corneal surface. P-cadherin (P-cad) is a critical stem cell niche adhesion molecule at various epithelial stem cell niches; however, conflicting observations were reported on the presence of P-cad in the limbal region. To explore this issue, we assessed the location and phenotype of P-cad^+^ cells by confocal microscopy of human corneoscleral tissue. In subsequent fluorescence-activated cell sorting (FACS) experiments, we used antibodies against P-cad along with CD90 and CD117 for the enrichment of LEPC, LMSC and LM, respectively. The sorted cells were characterized by immunophenotyping and the repopulation of decellularized limbal scaffolds was evaluated. Our findings demonstrate that P-cad is expressed by epithelial progenitor cells as well as melanocytes in the human limbal epithelial stem cell niche. The modified flow sorting addressing P-cad as well as CD90 and CD117 yielded enriched LEPC (CD90^−^CD117^−^P-cad^+^) and pure populations of LMSC (CD90^+^CD117^−^P-cad^−^) and LM (CD90^−^CD117^+^P-cad^+^). The enriched LEPC showed the expression of epithelial progenitor markers and better colony-forming ability than their P-cad^−^ counterparts. The cultured LEPC and LM exhibited P-cad expression at intercellular junctions and successfully repopulated decellularized limbal scaffolds. These data suggest that P-cad is a critical cell–cell adhesion molecule, connecting LEPC and LM, which may play an important role in the long-term maintenance of LEPC at the limbal stem cell niche; moreover, these findings led to further improvement of cell enrichment protocols to enhance the yield of LEPC.

## 1. Introduction

The integrity and function of corneal epithelial cells are maintained by limbal epithelial stem/progenitor cells (LEPC) located at a specific anatomic location referred to as the limbal stem cell niche [1]. The limbal stem cell niche is a specialized and complex microenvironment, where LEPCs are accompanied by non-epithelial supporting niche cells, blood vessels, and nerves in a specialized extracellular matrix (ECM) [2,3]. Interactions between limbal stem cells and supporting niche cells, which include limbal melanocytes (LM) and the limbal mesenchymal stromal cells (LMSC), are essential for the establishment and maintenance of niche architecture and for the transmission of regulatory signals to control cell division in limbal stem cell niche [4,5,6]. Several members of the cadherin family, including epithelial (E)-cadherin (-cad), neural (N)-cad, and placental (P)-cad, have been shown to mediate stem cell-niche interactions in epidermal, neural, mammary, and hematopoietic stem cell niches including the limbal stem cell niche [7,8,9,10,11,12]. While E-cad and N-cad have been extensively studied [10,11,13], less is known about the presence and function of P-cad in the limbal stem cell niche.

P-cad, a calcium-dependent cell–cell adhesion glycoprotein with the homophilic binding function was shown to exert a crucial role in the conservation of the structural integrity of epithelial tissues and it regulates processes involved in embryonic development, and maintenance of adult tissue architecture and cell differentiation [13,14,15]. P-cad expression has been reported in embryonic stem cells and many epithelial stem cell niches, including the limbal stem cell niche [13,16,17,18]; it has also been used to isolate bovine mammary epithelial stem cells [19]. P-cad expression was reported in the basal and immediate parabasal limbal epithelium of both adult and fetal corneas [18]; however, Hayashi and co-workers [10] reported that P-cad expression was not detected in the limbal or corneal region. Our earlier work suggested P-cad expression in melanocytes and basal limbal epithelial cells [12]; however, a detailed study of P-cad in the limbal stem cell niche is needed to clarify this issue, and moreover, an investigation on a possible use of P-cad as a selective marker to isolate LEPC has not been reported.

The successful enrichment of LEPC has been achieved by fluorescence-activated cell sorting (FACS) based on the expression of N-cad [10], a combination of integrin alpha 6 and CD71 [20], stage-specific embryonic antigen-4 [21], Hoechst dye efflux ability [22], CD200 [23], ATP-binding cassette sub-family 5 [24]; however, simultaneous enrichment of LEPC and niche cells from single preparations would allow the elucidation of limbal niche cell interactions. N-cad has been reported to enrich LEPC and LM from limbal tissue, but only a small fraction of LM express N-cad [10]. In a previous publication, we characterized CD90 and CD117 as selective markers to obtain pure populations of LMSC, LM, and limbal epithelial cells [25]; however, the limbal epithelial cells retrieved in this manner are a mixture of both progenitor cells (LEPC) and differentiated epithelial cells and LEPC are further enriched by culturing using cell-type-specific media [25]. Therefore, the addition of an epithelial progenitor marker to our earlier published protocol would allow the instant isolation and enrichment of LEPC along with a pure population of LMSC and LM.

In this study, we initially screened for cellular identity, location, and phenotype of P-cad^+^ cells by confocal microscopy at the limbal stem cell niche. LEPC were enriched using P-cad as a selective marker along with isolation of pure populations of limbal niche cells using CD90 and CD117 markers. The sorted limbal epithelial cells were characterized based on the expression of established LEPC markers at the cellular and molecular levels. The functional characteristics of LEPC and their ability to repopulate decellularized human limbal tissue was also analyzed in this study.

## 2. Materials and Methods

Human donor corneoscleral tissues (*n* = 3 for immunohistochemistry) are not suitable for transplantation and organ cultured corneoscleral tissue (*n* = 122 for cell isolation; *n* = 3 for immunohistochemistry) after retrieval of corneal endothelial transplants, with appropriate research consent provided by the Lions Cornea Bank Baden–Württemberg as described previously [25]. Informed consent to corneal tissue donation had been obtained from the donors or their relatives. The Institutional Review Board of the Medical Faculty of the University of Freiburg (25/20) approved experiments using human tissues that adhered to the Declaration of Helsinki.

### 2.1. Cell Isolation

Limbal cells were isolated, as previously described [26]. Briefly, organ-cultured corneoscleral tissue (mean age 71.7 ± 10.4 years; culture duration 28.0 ± 4.3 days; post mortem duration 29.3 ± 14.7 h; lightly pigmented donor limbal tissue; Appendix A) was cut into 12 three-clock-hour sectors, from which limbal segments were obtained by incisions made at 1 mm before and beyond the anatomical limbus. Limbal segments were enzymatically digested with collagenase A (Sigma-Aldrich, Roche; Mannheim, Germany; 2 mg/mL) at 37 °C for 18 h to generate clusters containing mixtures of epithelial, mesenchymal, and melanocytic cells. Cell clusters were separated from single cells by using reversible cell strainers with a pore size of 37 µm (Stem Cell Technologies, Köln, Germany). Subsequently, the cell clusters were dissociated into single cells with 0.25% Trypsin (Gibco, Life Technologies, ThermoFisher Scientific, Karlsruhe, Germany) containing 1 mM calcium chloride (PromoCell, Heidelberg, Germany) at 37 °C for 15–20 min, and the obtained single cells from pooled corneoscleral tissues (4–6 cornea in single preparation) were used for further processing.

Limbal cell clusters derived from collagenase digestion and single cells extracted after subsequent trypsin digestion were cultured in 4-well chamber slides for a week in corneal culture medium (CCM) containing Dulbecco’s modified Eagle medium/Ham’s F12 (3:1) (Hyclone; GE Healthcare Life Sciences, Freiburg, Germany) supplemented with bovine pituitary extract (BPE, 25 μg/mL), epidermal growth factor (EGF, 0.15 ng/mL) (Life Technologies, Carlsbad, CA, USA), 5% fetal calf serum (GE Healthcare Life Sciences), 0.4 mM calcium chloride (PromoCell) penicillin (100 U/mL)-streptomycin (100 μg/mL) mix (Gibco, Life Technologies, ThermoFisher Scientific, Karlsruhe, Germany) and Keratinocyte serum-free medium (KSFM; 0.08 mM Ca^2+^) supplemented with bovine pituitary extract, epidermal growth factor (Life Technologies, ThermoFisher Scientific, Karlsruhe, Germany), respectively. After 5 days of culture, the samples were processed for immunocytochemistry, as described below.

### 2.2. Fluorescence-Activated Cell Sorting (FACS)

FACS analysis was carried out as described previously [25]. Briefly, single-cell suspensions were incubated with FcR blocking reagent (Miltenyi Biotec, Bergisch Gladbach, Germany; 20 μL/10^6^ cells) for 5 min. Following washing, cells were incubated with a mouse anti-human CD117-PE, CD90-APC, and P-cad-Alexafluor-488 antibodies (5 μL/10^6^ cells) and their respective isotype controls ( Appendix A) in 100 µL phosphate-buffered saline (PBS), 0.1% sodium azide and 2% fetal calf serum for 40 min at 4 °C in the dark. Following cell washing, FACS was performed using a FACS Aria II sorter (BD Biosciences, Heidelberg, Germany) and the FACSDiva software (BD FACSDiva 8.0.1; BD Pharmingen; BD Biosciences). FlowJo software (FlowJo 10.2; Tree Star, Inc., Ashland, OR, USA) was used to analyze the post-acquisition data. The sorted cells were further processed for cytospin, realtime polymerase chain reaction (PCR), colony-forming unit (CFU) assays, and repopulation of scaffolds, as described below.

The sorted cells were adjusted to a density of 2 × 10^5^ cells/mL, after which 200 µL of the cell suspension was centrifuged at 1000 RPM for 5 min in a Cytospin 3 centrifuge (Themoshandon, Labstuff, Malente, Germany). The cytospin samples were further processed for immunocytochemistry as described below.

The CD90^−^CD117^+^P-cad^+^ sorted cells were seeded in LN-511-E8 (iMatrix-511, Nippo; 0.5 µg/cm^2^) coated T75 flasks (Corning, Tewksbury, MA, USA) and cultured in CNT-40 medium (CellnTec, Bern, Switzerland) at 37 °C, 5% CO_2_ and 95% humidity. CD90^−^CD117^−^/P-cad^+^ and CD90^−^CD117^−^/P-cad^−^ cells were seeded on 3T3 fibroblasts as mentioned below for colony-forming unit or seeded into T75 flasks in KSFM. CD90^+^CD117^−^/P-cad^−^ were seeded on a T75 flask in Mesencult media (Stem Cell Technologies). All the cultures were incubated at 37 °C, 5% CO_2_ and 95% humidity, and the media was changed every other day.

For immunostaining purposes, the passage 1 cells of CD90^−^CD117^−^/P-cad^+^ epithelial cells and CD90^−^CD117^+^/P-cad^+^ LM were cultured in 4-well chambers in KSFM and CNT-40 media respectively. After the confluence of CD90^−^CD117^−^/P-cad^+^ cells, the medium was shifted to CCM and cultured for 48 h, to promote the formation of adhesion molecules.

### 2.3. CFU-E Assay

Clonal expansion of both CD90^−^CD117^−^/P-cad^+^ and CD90^−^CD117^−^/P-cad^−^ cells were studied on feeder layers using mitomycin C-treated 3T3 fibroblasts as described previously [27]. Briefly, the sorted cells were seeded at a density of 300 cells/cm^2^ on the feeder layer. After 14 days of culture in CCM, the colonies were stained using 0.5% crystal violet. The colony-forming efficiency (CFE) was calculated using the formula: number of colonies formed/number of cells plated × 100% as d and the colony growth area was calculated as colony growth area/total culture area × 100%. For colony counting, holoclones, meroclones, and paraclones were included in the counting.

### 2.4. Co-cultures

To study the role of P-cad in heterotypic cell–cell interactions, the cultured CD90^−^CD117^−^/P-cad^+^ (P1) and CD90^−^CD117^+^/P-cad^+^ LM (P1) were seeded at equal density (5000 cells/well each) on 4-well chamber slides and cultured for 10 days in CCM media. The cultured cells were processed for immunocytochemistry as described below.

### 2.5. Real-time RT-PCR

RNA isolation, first-strand cDNA synthesis and PCR reactions were performed as previously described [28]. Briefly, RNA was isolated from sorted both CD117^−^CD90^−^P-cad^+^ and CD117^−^CD90^−^P-cad^−^ cells using the RNeasy Micro Kit (Qiagen, Hilden, Germany) and first-strand cDNA synthesis was performed using 2 µg of RNA and Superscript II reverse transcriptase (Invitrogen, Karlsruhe, Germany). PCR reactions were run in triplicate using TaqMan Probe Mastermix (Roche Diagnostics, Mannheim, Germany) and the comparative *C*_T_ method (ΔΔCT) was used to normalize gene expression levels relative to the housekeeping gene GAPDH. A gene was considered differentially expressed when its expression levels exceeded a two-fold difference across all specimens analyzed (*n* = 3). Primer sequences (Sigma-Aldrich) are given in Appendix A.

### 2.6. Flow Cytometry

The cultured CD117^−^CD90^−^P-cad^+^ LEPC (P1) and CD117^−^CD90^−^P-cad^+^ LM (P1) were trypsinized using 0.25% Trypsin in presence of 1 mM CaCl_2_ at 37°C for 5 min and the trypsin action was inhibited by using DMEM containing 10% fetal bovine serum. Flow cytometry was carried out as described previously [26]. Briefly, single-cell suspensions (0.5–1 × 10^6^ cells) were incubated with P-cad-APC and a respective isotype control. After three washes, cells were resuspended in ice-cold PBS, and flow cytometry was performed on a FACSCanto II (BD Biosciences) by using FACS Diva Software (BD FACSDiva 8.0.1; https://www.bdbiosciences.com/en-us/instruments/research-instruments/research-software/flow-cytometry-acquisition/facsdiva-software, accessed on 12 June 2022). A total of 10,000 events were acquired. A post-acquisition analysis was conducted using the FlowJo software (FlowJo 10.2, Tree Star Inc., Ashland, OR, USA, https://www.flowjo.com/, accessed on 12 June 2022).

### 2.7. Immunohistochemistry of Frozen Sections and Immunocytochemistry

Corneoscleral tissue samples (mean age 75.2 ± 10.9 years) within 16 h after death and organ-cultured corneoscleral tissue samples (mean age 58.3 ± 1.1, post mortem duration 14.1 ± 5.1 h; culture duration 34.8 ± 1.2 d) were embedded in an optimal cutting temperature (OCT) compound and frozen in liquid nitrogen. Cryosections of 6 μm thickness were cut from the superior or inferior quadrants, cultured cells on 4 well-glass chamber slides (LabTek; Nunc, Wiesbaden, Germany) and cytospin preparations were fixed in 4% paraformaldehyde for 15 min, blocked with 10% normal goat serum (NGS) and incubated in primary antibodies ( Appendix A) diluted in 2% NGS, 0.1% Triton X-100 in PBS overnight at 4 °C or 3 h at room temperature. Antibody binding was detected by Alexa-488-,-568-,647-conjugated secondary antibodies (Life Technologies, Carlsbad, CA, USA) and mounted in Vectashield antifade mounting media with DAPI (Vector, Burlingame, CA, USA). A laser scanning confocal microscope (TCS SP-8, Leica, Wetzlar, Germany) was used to examine immunolabelled samples. For negative controls, the primary antibodies were replaced by PBS.

### 2.8. Histology and Immunohistochemistry—Paraffin

For routine histology, the scaffolds were fixed in 4% paraformaldehyde for 30 min and embedded in paraffin. The 5 µm thick sections were cut and stained with hematoxylin (Haematoxylin Gill III, Surgipath, Leica, Germany) and eosin Y (Surgipath, Leica, Germany) as described previously [29].

Immunohistochemistry was performed as previously described [30]. The list of antibodies is provided in Appendix A.

### 2.9. Recellularization of Decellularized Human Limbal Tissue

Decellularized human limbal scaffolds (DHL) were prepared as described previously [29]. Recellularization of decellularized scaffolds was also carried out as described previously [29]. Briefly, for DHL-LEPC/LM scaffolds, the cultured LEPC (P1) and LM (P1) were seeded together in a ratio of 3:1 on the decellularized corneal surface and cultured in CCM (0.4 mM Ca^2+^). For stratification of scaffolds after 1 week, the tissues were raised to the air–liquid interface and the culture conditions shifted to high-calcium concentrations (1.2 mM Ca^2+^) and cultured for a further 2 weeks. All cultures were maintained at 37 °C, 5% CO_2_, and 95% humidity, and the medium was changed every alternative day. After terminating the cultivation, limbal scaffolds were fixed for immunohistochemistry and light microscopy as described above.

### 2.10. Statistical Analysis

The statistical analyses were performed as described earlier [25]. Briefly, the GraphPad InStat statistical package for Windows (Version 6.0; Graphpad Software Inc., La Jolla, CA, USA; https://www.graphpad.com/, accessed on 12 June 2022) was used to perform statistical analyses. Results are expressed as mean ± standard error of the mean (SEM) from individual experiments. The statistical significance (*p* value < 0.05) was determined with the Mann–Whitney U test.

## 3. Results

### 3.1. Localization of P-cad at the Corneal Limbus

Immunohistochemical staining of fresh corneoscleral tissues (non-cultured) showed the expression of P-cad (green) in the basal layers of limbal epithelium and a rather weak expression in the corneal basal epithelium (dashed line represents basement membrane (BM); Figure 1A). Double immunostaining confirmed the colocalization of epithelial keratins (pan-cytokeratin (CK), red) and P-cad (green) in the basal limbal epithelium (dashed line represents BM, Figure 1B); colocalization of melan-A (red) and p-cad (green) in the melanocytes (arrowheads); whereas sub-epithelial stromal cells (vimentin^+^, red) were in close association with P-cad^+^ limbal basal epithelial cells (green, Figure 1B); these data suggest that P-cad is present in basal limbal epithelial cells as well as limbal melanocytes. Double immunostaining of limbal tissue revealed that P-cad^+^ cells (green) were not co-localized with epithelial differentiation markers CK3 and CK12 (red), but it co-localized with progenitor/stem cell markers CK14, CK15, N-cad (arrowheads) and p63 (red, dashed line represents BM) (Figure 1C); this further suggests that P-cad expression is associated with LEPC but not with differentiated epithelial cells. At the basal layer of limbal epithelium, P-cad^+^ cells (green) did not show the expression (or rather a weak expression) of E-cad (red, arrowheads), whereas co-localization of P-cad^+^ cells with E-cad was observed in basal corneal epithelium (dashed line represents BM; Figure 2A).

Immunostaining on organ-cultured corneoscleral tissue was also performed to see the effect of culture conditions on P-cad expression. Similar to fresh limbal tissues, immunohistochemical staining showed the colocalization of P-cad^+^ cells with CK15 and Melan-A (dashed line represents BM, Figure 2B); however, the quality of organ-cultured limbal tissue in terms of epithelial layers was reduced (especially superficial layers) as indicated by a lower number of P-cad^−^ epithelial cells compared to fresh limbal tissue (dotted line separates P-cad^+^ and P-cad^−^ cells, Figure 2B).

Triple immunostaining of cultured limbal clusters (Figure 2C, the dashed line indicates the edge of the colony), derived from collagenase digestion, and cultured single cells (Figure 2D), derived from trypsin dissociation of clusters showed the presence of melanocytes (melan-A^+^ (red)vimetin^+^(cyan); arrow) and stromal cells (vimentin^+^(cyan); arrowheads) in limbal epithelial cell (pan-CK^+^ (green)vimentin^+^ (cyan)) cultures; these data suggest contamination of LM and LMSC in unsorted epithelial cultures.

### 3.2. Flow Sorting and Characterization of Limbal Cells

The limbal cell suspensions were gated to select cells of interest and to enrich single cells followed by dead cell exclusion using 4′,6-diamidino-2-phenylindole (DAPI) as described earlier [25]. Then, gates were set based on isotype controls to select CD90^+^, CD90^−^ (Figure 3A(i)) cells initially and later for CD90^−^CD117^−^P-cad^+^; CD117^+^CD90^−^P-cad^+^; CD117^−^CD90^−^P-cad^−^ cells (Figure 3A(ii)). Limbal cluster-derived cell suspensions from donor corneal samples provided a yield of 2.9 ± 0.9% of CD90^+^ cells (700–3888/limbus); 1.2 ± 0.4% of CD90^−^CD117^+^P-cad^+^ cells (250–857/limbus); 33.7 ± 9.8% of CD90^−^CD117^−^P-cad^−^ cells (11,400–36,428/limbus); and 52.4 ± 12.4% of CD90^−^CD117^−^P-cad^+^ cells (14,400–64,285/limbus) (Figure 3A(iii,iv),B,C).

The cytospin preparations of sorted CD90^+^ cells exhibited a pan-CK^−^/Vimentin^+^/Melan-A^−^ phenotype on immunostaining, a characteristic feature of LMSC (Figure 3D). CD90^−^CD117^+^P-cad^+^ cells stained for pan-CK^−^/Vimentin^+^/Melan-A^+^, a characteristic feature of melanocytes. CD90^−^CD117^−^P-cad^+^ cells exhibited a pan-CK^+^/Vimentin^+^/Melan-A^−^ phenotype, which is characteristic of LEPC, whereas CD90^−^CD117^−^P-cad^−^ cells had a pan-CK^+^/Vimentin^−^/Melan-A^−^ phenotype with relatively large cells, characteristic of differentiated epithelial cells (Figure 3D).

The cultural and functional characteristics of CD90^+^ LMSC and CD90^−^CD117^+^P-cad^+^ LM are similar to those described previously [25].

### 3.3. Progenitor Cell Properties of CD90^−^CD117^−^P-cad^+^ and CD90^−^CD117^−^P-cad^−^ Limbal Epithelial Cells

To assess the stem/progenitor cell properties of FACS sorted CD90^−^CD117^−^P-cad^+^ and CD90^−^CD117^−^P-cad^−^ limbal epithelial cells, quantitative real-time polymerase chain reaction (qRT-PCR), immunophenotyping, in vitro clonogenicity and self-renewal assays were performed. The gene expression analysis of sorted CD90^−^CD117^−^P-cad^+^ cells showed a significantly higher expression of the epithelial progenitor maker (keratin (KRT)15, 2.6 ± 0.2-folds; *p* = 0.02) and lower expression of differentiated marker (KRT3, 6.7 ± 0.8-folds; *p* = 0.02) compared to CD90^−^CD117^−^P-cad^−^ cells (Figure 4A). No significant difference was observed for epithelial markers (CDH1 (E-cad), integrin α3 (ITGA3)) between the samples (Figure 4A). The progenitor marker Ki-67 (MKI27) was strongly expressed in CD90^−^CD117^−^P-cad^+^ cells compared to CD90^−^CD117^−^P-cad^−^ cells (2.2 ± 0.1-fold; *p* = 0.02, Figure 4A). Double immunostaining of sorted CD90^−^CD117^−^P-cad^+^ cells revealed the expression of progenitor markers CK15, p63, and CK19 (~95% cells) and only very few cells expressed CK3 (~2%), whereas ~96% of CD90^−^CD117^−^P-cad^−^ cells expressed CK3 and only a few cells (~3%) expressed CK15 and CK19 (Figure 4B).

We also assessed self-renewal as a key property of stem/progenitor cells, by measuring the cell numbers after 10 days in culture (phase-contrast micrographs, Figure 4C). The CD90^−^CD117^−^P-cad^+^ cells showed significantly higher proliferative potential compared to CD90^−^CD117^−^P-cad^−^ cells (2.9-fold; Figure 4C). Colony-forming assays were used to investigate the enrichment of functional epithelial progenitors in FACS-sorted CD90^−^CD117^−^/P-cad^+^ and CD90^−^CD117^−^/P-cad^−^ cell fractions. Both cell fractions formed colonies; however, CD90^−^CD117^−^P-cad^+^ cells showed a higher CFU efficiency (1.2% of CD90^−^CD117^−^P-cad^+^ vs 0.2% of CD90^−^CD117^−^P-cad^−^; *p* = 0.001) and a larger growth area (41.8% of CD90^−^CD117^−^P-cad^+^ vs. 9.3% of CD90^−^CD117^−^P-cad^−^, *p* = 0.03) (Figure 4D), suggesting an enrichment of clonogenic epithelial cells in the CD90^−^CD117^−^P-cad^+^ population.

### 3.4. P-cad Is Associated with LEPC and LM

P-cad expression was evaluated in cultured CD90^−^CD117^−^P-cad^+^ LEPC (P1) and CD90^−^CD117^+^P-cad^+^ LM (P1) by immunophenotyping. Flow cytometric analysis showed P-cad expression on cultured CD90^−^CD117^−^P-cad^+^ (96.2 ± 2.4%) and CD90^−^CD117^+^P-cad^+^ (98.7 ± 1.6%) cells (Figure 5A). Immunostaining also showed membranous staining of P-cad in CD90^−^CD117^−^P-cad^+^ cells cultured in 0.08 mM Ca^2+^ and its enrichment at cell–cell junctions in 1.2 mM Ca^2+^ (Figure 5B). Double immunostaining of CD90^−^CD117^+^P-cad^+^ LM revealed the colocalization of Melan-A and P-cad, which suggests P-cad-mediated homophilic interaction with LM (Figure 5B). To further characterize the role of P-cad as a cell–cell adhesion molecule at the limbal stem cell niche, immunostaining was performed on limbal clusters and co-cultures of CD90^−^CD117^−^P-cad^+^ LEPC and CD90^−^CD117^+^P-cad^+^ LM. Immunostaining of limbal clusters showed expression of P-cad in both LEPC as well as LM, which are interspersed between basal epithelial cells (Figure 5C). In higher magnification, signal enhancement at sites of cell–cell overlap is compatible with P-cad-mediated adherence of CD90^−^CD117^+^P-cad^+^ LM to CD90^−^CD117^−^P-cad^+^ LEPC (Figure 5C). Similar observations were made using in vitro co-cultures of LEPC and LM (Figure 5D).

### 3.5. Repopulation of Decellularized Limbal Scaffolds

The repopulation potential of sorted CD90^−^CD117^−^P-cad^+^ LEPC was tested by seeding these cells on DHLscaffolds; we also used CD90^−^CD117^+^P-cad^+^ LM as they share a common locality with LEPC in vivo. Hematoxylin and Eosin (H&E) staining of the DHL specimen showed a regular arrangement of collagen fibrils, connective tissue protrusions (black arrows), invaginations (white arrows), and vascular gaps (dashed circles) in the ECM (Figure 5E(i)) [29]. After seeding of both cell types on DHL scaffolds and three weeks of cultivation, stratification (2–3 layers) of the epithelium, and darkly pigmented cells interspersed between the epithelial cells (supposed to be melanocytes) were confirmed by H&E staining (the dotted line represents the BM; Figure 5E(ii)). We also observed a repopulation of the limbal stroma (Figure 5E(ii)) similar to earlier observations [29].

Phenotypic characterization of recellularized limbal scaffolds by immunohistochemical staining confirmed pronounced epithelial keratins (pan-CK) expression and intercellular E-cad in all epithelial layers (Figure 5F); vimentin staining was observed in basal layers and also in the limbal stroma (Figure 5F); Melan-A^+^ melanocytes were interspersed in the epithelial layers (arrowhead, Figure 5F), (dashed line represents BM); CK15 and p63 staining (arrowheads) were detected in the basal layer (dashed line separates basal and suprabasal cells, Figure 5F).

## 4. Discussion

Cadherins are cell–cell adhesion molecules that have been shown to be involved in cell adhesion in various stem cell niches including the limbal stem cell niche [7,8,9,11,12,20] and contradictory results have been reported on the expression of P-cad at the limbal stem cell niche [10,18]. Therefore, a detailed study of P-cad at the limbal stem cell niche is warranted to clarify this issue and improve our understanding of the role of P-cad. In the present study, P-cad expression was observed in epithelial and melanocytes of basal limbus suggesting P-cad is a mediator of the intercellular interactions of basal limbal epithelial cells similar to N-cad [10]. The current study also illustrates a co-expression of P-cad with progenitor cell markers CK15, N-cad, CK14, p63α, but not with CK3, CK12, E-cad, known markers of epithelial differentiation [9]; these data suggest that P-cad can be used as a marker for the enrichment of LEPC similar to N-cad [10]. The expression of P-cad by melanocytes in the limbal stem cell niche suggests that melanocytes have direct P-cad-mediated contact with LEPC. Interestingly, we also observed a rather weak expression of P-cad in corneal basal epithelial cells, which strongly express E-cad, whereas P-cad and E-cad appeared mutually exclusive at the limbal basal layer; this suggests that cells migrating into the suprabasal compartments or towards the cornea down-regulate P-cad expression and upregulate E-cad, as has been described in the epidermal stem cell compartment [31,32]. Previous studies have reported a prominent cell loss and down-regulation of putative stem cell markers in organ-cultured limbal tissue [33]. Similarly, a loss of superficial cells was observed in the organ-cultured corneas used in this study, which is reflected in the lower number of P-cad^−^ cells compared to P-cad^+^ cells. Therefore, several factors influence the viability and differentiation status of LEPC such as donor cornea death-to-preservation time, storage procedure, and donor age, which may act as limitations of our study.

LEPC have been enriched by flow sorting based on the expression of various cell surface markers, including the cell–cell adhesion molecule N-cad [10,21,22,23,24]. All these protocols successfully enriched LEPC, but the simultaneous enrichment of LEPC along with isolation of limbal niche cells was lacking. N-cad has been reported to enrich LEPC and melanocytes from limbal tissue, but only a small fraction of LM expressed N-cad [10]. In our previous publication, we have successfully isolated pure populations of LM, LMSC and limbal epithelial cells [25]. The isolated limbal epithelial cell fractions obtained in our previous study were devoid of both LM and LMSC (CD90^−^CD117^−^) and contained all limbal epithelial cells, including both progenitors (LEPC) and differentiated cells [25]. To enhance the purity of the LEPC fraction, a P-cad was included as an additional separation marker in the present study. Our findings suggest successful isolation and pure populations of LMSC (CD90^+^CD117^−^), LM (CD90^−^CD117^−^P-cad^+^) and limbal epithelial cells (both CD90^−^CD117^−^P-cad^+^ and CD90^−^CD117^−^P-cad^−^ cells) by FACS. Among the limbal epithelial fractions, CD90^−^CD117^−^P-cad^+^ cells are smaller in size compared to CD90^−^CD117^−^P-cad^−^ cells and also expressed vimentin suggesting a more immature progenitor cell phenotype [9,10]. To further confirm the possibility that CD90^−^CD117^−^P-cad^+^ fractions were enriched for epithelial stem/progenitor cells, they were subjected to immunophenotyping, gene expression analysis, and functional CFU assays. The CD90^−^CD117^−^P-cad^+^ cells showed significantly higher expression of stem/progenitor markers (CK15, CK19, CK14, and p63) and reduced expression of differentiated corneal epithelial markers (CK3) on either mRNA or protein levels when compared to CD90^−^CD117^−^P-cad^−^ cells. Similarly, our data also confirm that CD90^−^CD117^−^P-cad^+^ cells have higher proliferation potential with better colony-forming capacity suggesting enrichment of stem/progenitor cells in the CD90^−^CD117^−^P-cad^+^ fraction; moreover, the enriched LEPC, LMSC, and LM fractions would allow the molecular and cellular characterization of limbal niche cells and their interactions at the limbal stem cell niche. Of note, the protocol described in this article may hold the possibility to serve as an alternative method to enrich epithelial progenitor cells, melanocytes, and niche fibroblasts of other epithelial stem cell niches.

Limbal melanocytes, which reside in association with LEPC in basal layers, are not only professional melanin-producing cells, but exert various non-canonical functions in limbal niche homeostasis by regulating LEPC maintenance, immune-response, and angiostasis [4,25,27,34]; however, the mechanism by which these functions are exerted is not yet elucidated. It has been reported that N-cad is involved in heterotypic interactions between LEPC and a fraction of LM [10]. On the contrary, Higa et al. reported that they were not able to find any evidence of N-cad-mediated adhesion between melanocytes and LEPC [11]. In the present study, we provide evidence for P-cad expression in LEPC and enrichment of LEPC and melanocytes in a P-cad^+^ cell fraction. These data strongly suggest that P-cad is not only involved in homotypic interactions of neighboring LEPC but also in heterotypic contacts between LEPC and melanocytes similar to observations in epidermal and hair follicle stem cell niches [31,35,36]. Therefore, homophilic P-cad-mediated adhesion of LEPC and LM may play an important role in the long-term maintenance of LEPC within the limbal stem cell niche. Further investigation of P-cad and its functional involvement in an LEPC-melanocyte co-culture model is required to fully elucidate the importance of these interactions and their role in the modulation of stem cell fate in native tissue.

Various biomaterials have been used to culture limbal epithelial cells for clinical transplantation [37,38]. Decellularized scaffolds have the unique advantage of a tissue-specific three-dimensional structure [39,40]. Recently, we have shown that decellularized limbal scaffolds provide a limbus-specific microenvironment and could be a promising scaffold to transplant LEPC for the treatment of LSCD [29,41]. In the present study, we have tested the repopulation capacity of CD90^−^CD117^−^P-cad^+^ LEPC and CD90^−^CD117^+^P-cad^+^ LM on decellularized limbal scaffolds. We observed that LEPC expanded on scaffolds and displayed a well-organized stratified structure as well as preservation of a progenitor cell phenotype in the basal layer with melanocytes present in the epithelial layers. These data strongly suggest that enriched CD90^−^CD117^−^P-cad^+^ LEPC could repopulate the decellularized scaffold, which could be used for transplantation in patients suffering from LSCD in the future.

In conclusion, we provide solid evidence for the presence of a P-cad at the human limbus based on studies ex vivo and in vitro. At the limbus, P-cad is localized to sites of epithelial progenitor cell interactions and the interaction of limbal melanocytes and epithelial progenitor cells in the basal cell layer. These findings led to a further improvement of cell enrichment protocols to enhance the yield of LEPC; moreover, homophilic P-cad-mediated adhesion between LEPC and LM may play an important role in the long-term maintenance of LEPC in the limbal stem cell niche.

## Figures and Tables

**Figure 1 cells-11-01975-f001:**
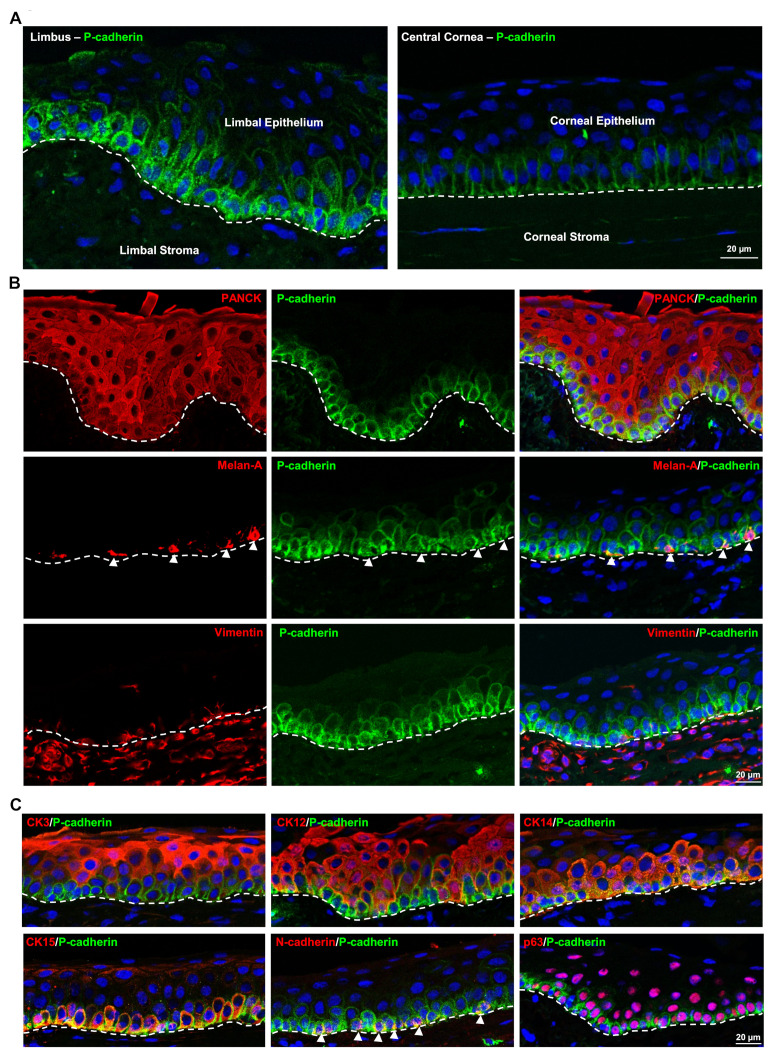
Localization of P-cadherin (P-cad) at the corneal limbus: (**A**) Immunohistochemical staining of fresh corneoscleral tissues showing the expression of P-cad (green) in the basal layers of limbal epithelium and a rather weak expression in the corneal basal epithelium (dashed line represents basement membrane (BM)). Nuclear counterstaining with 4′,6-diamidino-2-phenylindole (DAPI, blue). (**B**) Double immunostaining analysis of limbal tissue sections showing the colocalization of epithelial keratins (pan-cytokeratin (CK), red) and P-cad (green) in the basal limbal epithelium); colocalization of melan-A (red) and P-cad (green) in the melanocytes (arrowheads); whereas sub-epithelial stromal cells (vimentin^+^, red) were in close association with P-cad^+^ limbal basal epithelial cells (green, dashed line represents BM). Nuclear counterstaining with DAPI (blue). (**C**) Double immunostaining of limbal tissue sections showing non-colocalization of P-cad^+^ cells (green) with epithelial differentiation markers CK3 and CK12 (red), whereas colocalization with progenitor/stem cell markers CK14, CK15, N-cadherin (arrowheads), and p63 (red, dashed line represents BM) (**C**). Nuclear counterstaining with DAPI (blue).

**Figure 2 cells-11-01975-f002:**
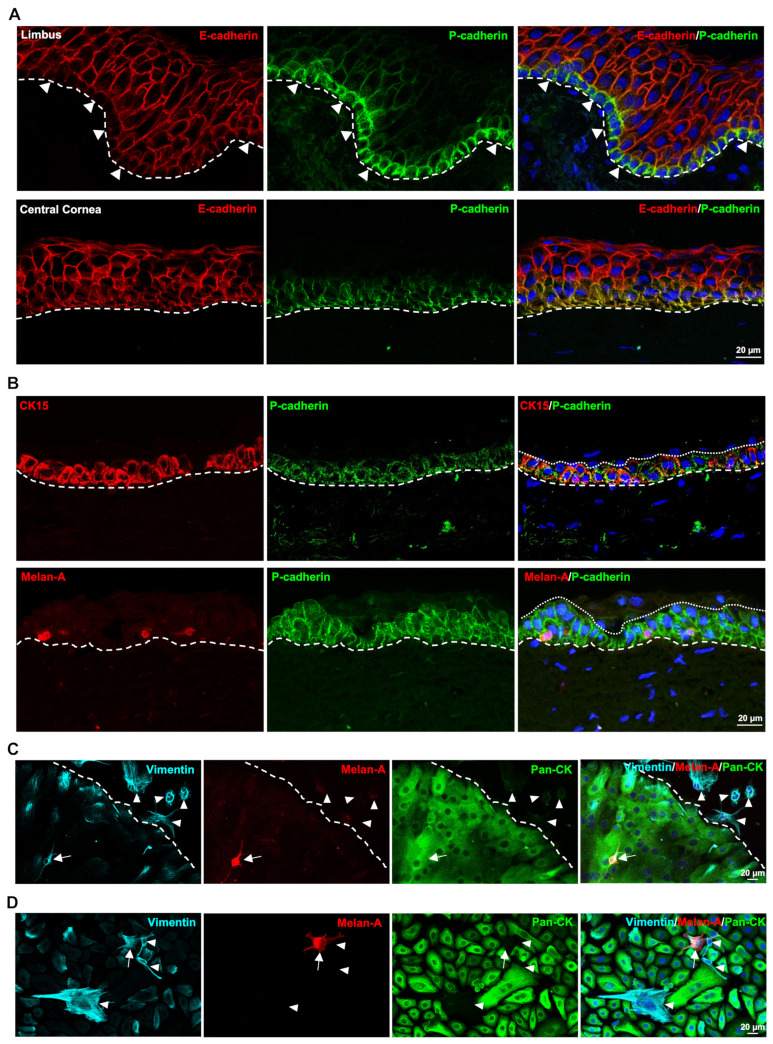
Characterization of the corneal limbus and unsorted cultures: (**A**) Immunohistochemical staining of fresh corneoscleral tissues showing non-colocalization of the P-cad^+^ cells (green) with E-cadherin (no expression or rather weak expression; red, arrowheads), whereas colocalization of P-cad^+^ cells with E-cadherin in basal corneal epithelium (dashed line represents the basement membrane (BM)). Nuclear counterstaining with 4′,6-diamidino-2-phenylindole (DAPI, blue). (**B**) Double immunostaining of organ-cultured corneoscleral tissues showing the co-localization of P-cad^+^ cells with CK15 and Melan-A (red, dashed line represents BM; dotted line separates P-cad^+^ and P-cad^−^ cells). Nuclear counterstaining with 4′,6-diamidino-2-phenylindole (DAPI, blue). (**C**,**D**) Triple immunostaining analysis of cultured limbal clusters (**C**, dashed line represents the edge of the epithelial colony) and cultured single cells (**D**) showing the presence of limbal epithelial cells (pan-CK^+^ (green)vimentin^+^ (cyan)), melanocytes (melan-A^+^ (red)vimetin^+^(cyan); arrow) and stromal cells (vimentin^+^(cyan); arrowheads). Nuclear counterstaining with DAPI (blue).

**Figure 3 cells-11-01975-f003:**
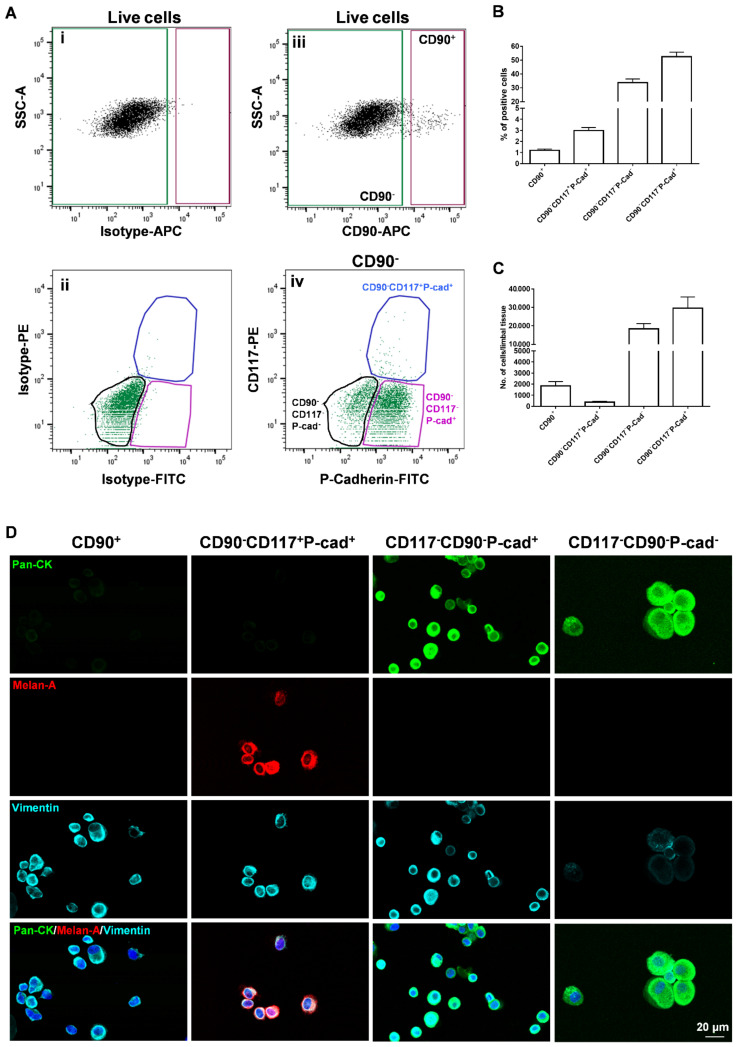
Flow sorting of limbal niche cells and characterization: (**A**) Fluorescence-activated cells sorting (FACS) images demonstrating the gating strategy used to isolate limbal cells. The isotype control graphs (i and ii) showing the set of gates to select CD90^+^, CD90^−^ (iii) cells initially and later for CD90^−^CD117^−^P-cad^+^; CD117^+^CD90^−^P-cad^+^; CD117^−^CD90^−^P-cad^−^ cells (iv). (**B**) The graph shows the percentage of CD90^+^CD117^−^, CD90^−^CD117^+^, CD90^−^CD117^−^ cells obtained from the limbus. Data are expressed as the means ± SEM of 25 individual experiments including 122 corneoscleral tissues. (**C**) The graph shows the number of CD90^+^, CD90^−^CD117^+^P-cad^+^, CD90^−^CD117^−^P-cad^+^ and CD90^−^CD117^−^P-cad^−^ cells obtained from the limbus. Data are expressed as the means ± SEM of 25 individual experiments including 122 corneoscleral tissues. (**D**) Triple immunostaining of cytospin preparations of sorted cells showing vimentin^+^ cells in CD90^+^ fraction; Melan-A (red) and vimentin (cyan) double-positive cells in CD90^−^CD117^+^P-cad^+^ fraction; CD90^−^CD117^−^ P-cad^+^ cells stained for pan-cytokeratin (PCK) and vimentin; CD90^−^CD117^−^P-cad^+^ cells stained for PCK. Nuclear counterstaining with 4′,6-diamidino-2-phenylindole (blue).

**Figure 4 cells-11-01975-f004:**
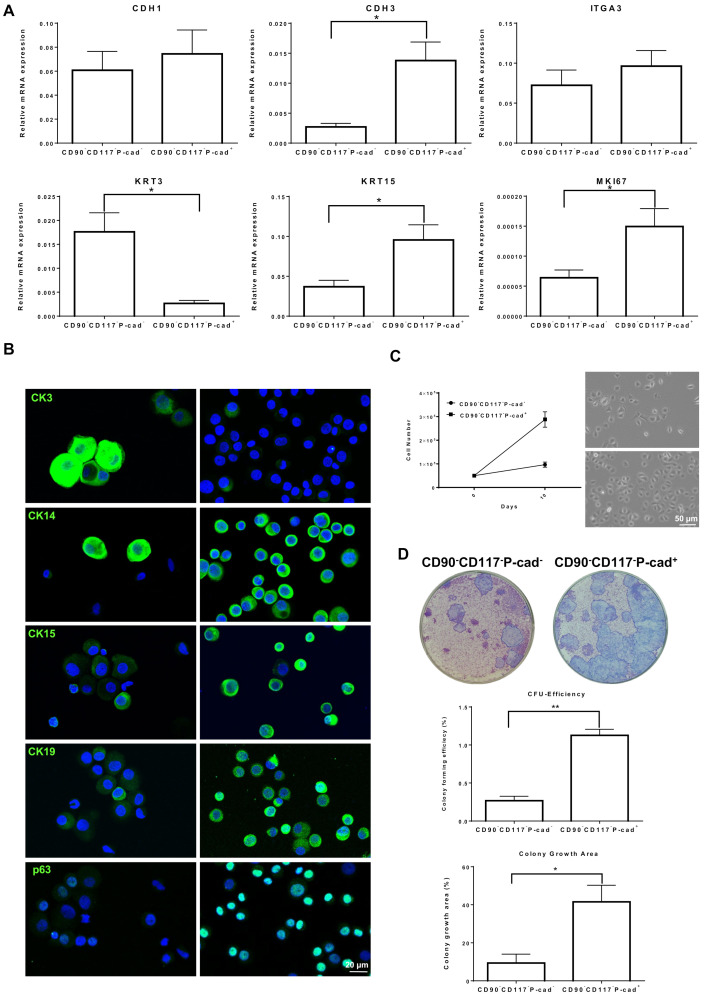
Characterization of P-cadherin^+^ (P-cad^+^) limbal epithelial cells: (**A**) Quantitative real-time polymerase chain reaction primer assays confirming the differential expression of limbal progenitor (Keratin (KRT)15), differentiated marker (KRT3) and proliferative cell marker (Ki-67 (MKI67) and similar expression of epithelial markers (E-cadherin (CDH1), integrin α3 (ITGA3)) in sorted CD90^−^CD117^−^P-cad^+^ and CD90^−^CD117^−^P-cad^−^ cells. Data are expressed as means (2−ΔCT)  ±  SEM (*n* = 3). * *p* < 0.05; Mann–Whitney U test. (**B**) Immunostaining of cytospin preparation of sorted cells confirming the differential expression of epithelial progenitor markers (cytokeratin (CK)15, p63, and CK19) and differentiated marker (CK3) (green) in CD90^−^CD117^−^P-cad^+^ and CD90^−^CD117^−^P-cad^−^ cells. Nuclear counterstaining with 4′,6-diamidino-2-phenylindole (blue). (**C**) Graph and phase-contrast micrographs confirming the differential proliferation of the potential of cultured CD90^−^CD117^−^P-cad^+^ and CD90^−^CD117^−^P-cad^−^ cells. Data are expressed as means ± SEM of 3 individual experiments. (**D**) The sorted CD90^−^CD117^−^P-cad^+^ and CD90^−^CD117^−^P-cad^−^ cells form typical cellular colonies on the NIH/3T3 fibroblast feeder layers after 14 days in culture. Colony-forming analysis showing significant higher colony-forming ability and growth covered area in CD90^−^CD117^−^P-cad^+^ than CD90^−^CD117^−^P-cad^−^ cells. Percentage of colony-forming efficiency and growth area are expressed as means ± standard error of the mean of 4 individual experiments. * *p* < 0.05; ** *p* < 0.01.

**Figure 5 cells-11-01975-f005:**
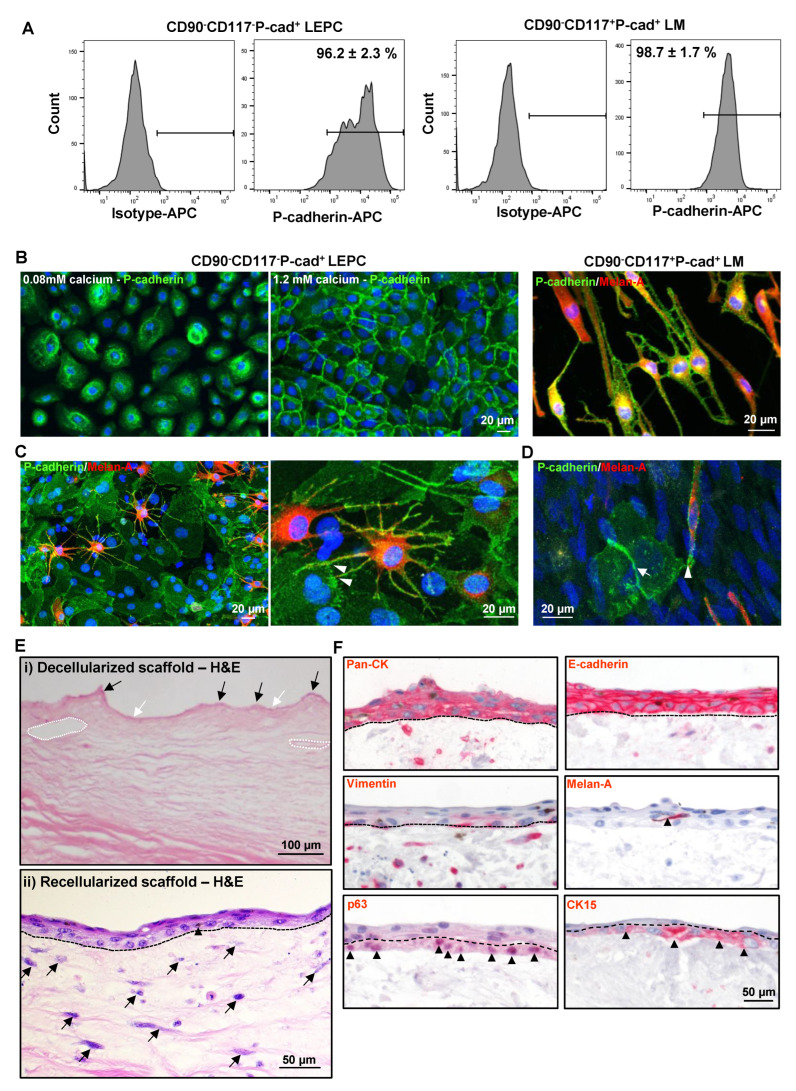
Association of P-cadherin in limbal cells and recellularization of scaffolds: (**A**) Flow cytometric analysis of cultured CD90^−^CD117^−^P-cad^+^ and CD90^−^CD117^−^P-cad^−^ cells showing expression of P-cadherin. Data are expressed as means ± standard error of the mean of 4 individual experiments. (**B**) Immunstaining analysis of cultured epithelial cells showing membranous staining of P-cad in CD90^−^CD117^−^P-cad^+^ cells cultured in 0.08 mM Ca^2+^ and its enrichment at cell–cell junctions in 1.2 mM Ca^2+^. Double immunostaining of CD90^−^CD117^+^P-cad^+^ LM showing the co-localization of Melan-A and P-cad. Nuclear counterstaining with 4′,6-diamidino-2-phenylindole (DAPI, blue). C&D Double immunostaining of cultured cluster cells. (**C**) and in vitro co-cultures of CD90^−^CD117^−^P-cad^+^ LEPC and CD90^−^CD117^+^P-cad^+^ LM. (**D**) showing expression of P-cad in both LEPC and LM and signal enhancement at the sides of cell–cell overlap (arrowheads showing overlaps of LEPC and LM, arrows showing the overlap of epithelial cells) Nuclear counterstaining with DAPI (blue). (**E**) Hematoxylin and Eosin (HE) staining of decellularized scaffold (i) showing a regular arrangement of collagen fibrils, connective tissue protrusions (black arrows), invaginations (white arrows), and vascular gaps (dashed circles) in the ECM. After seeding of both LEPC and LM on DHL scaffolds and three weeks of cultivation, recellularized scaffolds (ii) showing the stratification (2–3 layers) of the epithelium, and darkly pigmented cells (arrowhead, supposed to be melanocytes) interspersed between the epithelial cells. The scaffolds also showing the repopulation of limbal stroma (arrows). (**F**) Immunohistochemical staining of recellularized limbal scaffolds showing pronounced epithelial keratins (pan-cytokeratin (pan-CK)) expression and intercellular E-cadherin in all epithelial layers; vimentin staining in basal layers and also in the limbal stroma; Melan-A^+^ melanocytes interspersed in the epithelial layers (arrowhead) (dashed line represents basement membrane); CK15, and p63 staining (arrowheads) in the basal layer (dashed line separates basal and suprabasal cells).

## Data Availability

The datasets generated during and /or analyzed during the current study are available from the corresponding author on reasonable request.

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
