# Peer review of "P-Cadherin Is Expressed by Epithelial Progenitor Cells and Melanocytes in the Human Corneal Limbus"

_cells, 2022, doi:10.3390/cells11121975_

Round 1

Reviewer 1 Report

The premise of interactions between LEPC and surrounding niche cells to play an important role in the maintenance of limbal stem cells are well received.  P-cad being a molecule that may support the above described interaction is also well received while the data have been conflicting.  The authors describe elegantly the presence and location of P-cad within the respective group of cells.  I was quite intrigued with the figures that very nicely describe the location of p-cad cells.  The figures look almost too perfect.  I would congratulate the authors for the experiments that they have conducted.  We hope that this discovery is repeated by independent labs.  The data does support their premise and hypothesis that P-cad plays an important role in maintenance of interaction of LEPC and surrounding niche cells

Reviewer 2 Report

Authors wrote an interesting article

I'd suggest to add the following paper in introduction:

1) PMCID: PMC7774169 DOI: 10.4103/ijo.IJO_2056_20

2) PMCID: PMC3818919 DOI: 10.1155/2013/857380

I'd suggest to increase the limitations of the study

Reviewer 3 Report

The manuscript by Polisetti et al. describes the expression of P-cadherin in limbal epithelial progenitor cells (LEPC) and limbal melanocytes and its role in long-term maintenance of limbal niche cells. It also describes a method to enrich LEPC using P-cadherin as a selective marker to isolate pure populations of limbal niche cells with CD90 and CD117 markers. The manuscript is well-written with interesting data. A few comments are below:

1. The main concern with the immunostaining results is the over-fixation of the corneal sections, which may make it seem like P-cadherin is only minimally expressed in the cornea.

2. The duration to obtain postmortem tissue seems rather long, especially for gene expression experiments.  

3. Its hard to tell which cell is a melanocyte in the H&E staining. Please explain.
